# Effects of Interspecific Chromosome Substitution in Upland Cotton on Cottonseed Macronutrients

**DOI:** 10.3390/plants10061158

**Published:** 2021-06-07

**Authors:** Nacer Bellaloui, Sukumar Saha, Jennifer L. Tonos, Jodi A. Scheffler, Johnie N. Jenkins, Jack C. McCarty, David M. Stelly

**Affiliations:** 1Crop Genetics Research Unit, USDA, Agriculture Research Service, 141 Experiment Station Road, Stoneville, MS 38776, USA; jennifer.tonos@usda.gov (J.L.T.); jodi.scheffler@usda.gov (J.A.S.); 2Genetics and Sustainable Agriculture Research Unit, USDA, Agriculture Research Service, Starkville, MS 39762, USA; sukumar.saha@usda.gov (S.S.); johnie.jenkins@usda.gov (J.N.J.); jack.mccarty@usda.gov (J.C.M.); 3Department of Soil and Crop Sciences, Texas A&M University, College Station, TX 77843, USA; stelly@tamu.edu

**Keywords:** cottonseed nutrition, macronutrients, chromosome substitution, cotton, mineral nutrition

## Abstract

Nutrients, including macronutrients such as Ca, P, K, and Mg, are essential for crop production and seed quality, and for human and animal nutrition and health. Macronutrient deficiencies in soil lead to poor crop nutritional qualities and a low level of macronutrients in cottonseed meal-based products, leading to malnutrition. Therefore, the discovery of novel germplasm with a high level of macronutrients or significant variability in the macronutrient content of crop seeds is critical. To our knowledge, there is no information available on the effects of chromosome or chromosome arm substitution on cottonseed macronutrient content. The objective of this study was to evaluate the effects of chromosome or chromosome arm substitution on the variability and content of the cottonseed macronutrients Ca, K, Mg, N, P, and S in chromosome substitution lines (CS). Nine chromosome substitution lines were grown in two-field experiments at two locations in 2013 in South Carolina, USA, and in 2014 in Mississippi, USA. The controls used were TM-1, the recurrent parent of the CS line, and the cultivar AM UA48. The results showed major variability in macronutrients among CS lines and between CS lines and controls. For example, in South Carolina, the mean values showed that five CS lines (CS-T02, CS-T04, CS-T08sh, CS-B02, and CS-B04) had higher Ca level in seed than controls. Ca levels in these CS lines varied from 1.88 to 2.63 g kg^−1^ compared with 1.81 and 1.72 g kg^−1^ for TM-1 and AMUA48, respectively, with CS-T04 having the highest Ca concentration. CS-M08sh exhibited the highest K concentration (14.50 g kg^−1^), an increase of 29% and 49% over TM-1 and AM UA48, respectively. Other CS lines had higher Mg, P, and S than the controls. A similar trend was found at the MS location. This research demonstrated that chromosome substitution resulted in higher seed macronutrients in some CS lines, and these CS lines with a higher content of macronutrients can be used as a genetic tool towards the identification of desired seed nutrition traits. Also, the CS lines with higher desired macronutrients can be used as parents to breed for improved nutritional quality in Upland cotton, *Gossypium hirsutum* L., through improvement by the interspecific introgression of desired seed nutrient traits such as Ca, K, P, S, and N. The positive and significant (*p* ≤ 0.0001) correlation of P with Ca, P with Mg, S with P, and S with N will aid in understanding the relationships between nutrients to improve the fertilizer management program and maintain higher cottonseed nutrient content.

## 1. Introduction

Macronutrients, including Ca, K, Mg, N, P, and K, are essential nutrients for crop growth, development, and seed production, including cottonseed. Macronutrients are involved in different physiological and metabolic processes. For example, K, Zn, and P are involved in CO_2_ fixation, photosynthesis, electron transfer, energy production, and protein and carbon-related enzymes [1,2]; P is involved in the photosynthesis process, carbon assimilation transport, and proteins and oils [2,3,4]; Mg, S, and B are involved in flower set, boll set, boll weight, and seed and cotton yield [3] due to the involvement of Mg, S, and B in metabolic activities, protein synthesis, and the mobilization of photosynthates [5,6]. Nutrients such as Mg and B are catalysts in enzymatic reactions that are required for respiration, meristematic development, chlorophyll formation, photosynthesis, protein and oil synthesis, gossypol metabolism, and tannin and phenolic metabolism [4,5,7,8]. Macronutrient levels in grain-based products are also important for human health, and deficiencies in some macronutrients in these products can lead to diseases and malnutrition. For example, the World Health Organization (WHO) reported that cardiovascular diseases (CVDs) are the leading cause of death globally, causing 17.8 million deaths in 2017, representing approximately 32% of all deaths worldwide [9]. In addition, high blood pressure (hypertension) was reported to be a major risk for CVDs. The same observation was made for children. It was also found that increased K intake significantly reduced blood pressure, stroke, and coronary heart disease in adults. The WHO recommended an increase in K intake from food to control blood pressure in children aged 2–15 years, and a K intake of at least 90 mmol/day for adults. Although the recommended level of intake of ≥90 mmol/day is still a conditional recommendation for adults due to limited research evidence of a precise level that results in maximum health benefits, consuming K at ≥90 mmol/day will provide health benefits. Another example is Ca and its role in hypertensive disorders where it can reduce the risk of pre-eclampsia and eclampsia, which are among the main causes of maternal deaths and preterm births, especially in low-income countries [10]. The survivors of births of this disease suffer from a higher risk of respiratory disease and long-term neurological morbidity. Ca supplementation improves calcium intake and consequently reduces the risk of hypertensive disorders during pregnancy. For low Ca diets, daily Ca supplementation (1.5–2.0 g oral elemental calcium) is recommended for pregnant women to reduce the risk of pre-eclampsia [10].

Nutrient accumulation in seeds is controlled by several processes, including nutrient uptake, translocation, redistribution, and accumulation [11,12]. Inheritance and genetic mechanisms of these processes need further research if ways to control these processes are to be identified [1,13]. The principal physiological and metabolic functions of plant macronutrients, including Ca, K, Mg, N, P, and S, are well documented [14,15], and quantitative trait loci (QTL) associated with mineral accumulation in seeds were identified for rice [16], wheat [17], and the alfalfa-related model organism *Medicago truncatula* [18]; limited QTLs were identified for the macronutrients Ca, Mg, K, N, S, and P in soybean seed [19]. While QTL mapping is a powerful tool for dissecting complexly inherited traits, such as seed mineral contents [20,21], and identifying genomic regions contributing to trait variation [22,23], it is typically ill-suited for fine mapping or gene cloning. Genetic association analyses of the target complex trait can enhance resolution and breeding selection efficiencies [24].

Molecular and genetic research on cottonseed macronutrients is needed. Chromosome substitution can be used to infuse new genetic variation into a crop species, some of which could significantly affect seed macronutrient contents. Since the genetics of cottonseed traits associated with chromosome substitution germplasm lines is scarce [25], the objective of the current research was to evaluate the effects of cotton chromosome substitution (CS) germplasm lines on the macronutrients Ca, K, Mg, P, N, C, and S in seeds of selected CS lines. The CS lines are near-isogenic to each other and the recurrent parent TM-1 with one chromosome or chromosome arm introgressed [26,27,28,29,30,31]. We used chromosome comparison (specific chromosomes or chromosome arms in the TM-1 background) to evaluate CS lines with differing macronutrient levels as a comparative method, which provided a tool that could be used for the improvement of Upland cotton seed nutritional quality [32].

## 2. Results

ANOVA analysis showed that Mg, P, C, N, and S were significantly (*p* < 0.0001) influenced by location, line, and their interactions (Table 1). The concentrations of Ca and K were only significantly (*p* < 0.0001) influenced by line, and location × line interactions. The effects of line, location, and their interactions on some macronutrients such as Mg, P, C, N, and S indicated the importance of the environmental factors of each location and the genetic makeup of each line on the accumulation of these nutrients in the cottonseed.

In South Carolina, the mean values showed that five CS lines (CS-T02, CS-T04, CS-T08sh, CS-B02, and CS-B04) had higher Ca levels in seed than both controls (Table 2). Ca levels in these CS lines ranged from 1.49 to 2.63 g kg^−1^ compared with 1.81 and 1.72 g kg^−1^ for TM-1 and AM UA48, respectively. CS T04 had the highest Ca concentration. CS M08sh exhibited the highest K concentration (14.50 g kg^−1^), a significant increase over TM-1 (Table 2). Lines CS-B02, CS-B04, CS-B08sh, CS-M02, CS-M04, CS-M08sh, CS-T02, and CS-T04 had higher Mg concentrations than both controls, with concentrations ranging from 2.73 to 2.85 g kg^−1^. Line CS-T04 had the highest concentration of Mg (2.85 g kg^−1^). Lines CS-M04 and CS-T04 had higher P concentrations than the controls and the rest of the CS lines, with CS-M04 being the highest (5.38 g kg^−1^). Both CS-M08sh and the control AMUA48 exhibited the highest concentration of C, but four CS lines (CS-B04, CS-M02, CS-T08sh, and CS-M08sh) were higher than TM-1. Seven CS lines were higher in S than the controls, with CS-T04 being the highest. Four (CS-B08sh, CS-T02, CS-T04, and CS-T08sh) out of nine lines had higher N than the controls, with CS-T02 having the highest concentration (47.79 g kg^−1^). Lines CS-B02, CS-B04, CS-B08sh, CS-M02, CS-T08sh, CS-T04, and CS-T02 had higher S than the controls, with CS-T04 being the highest (5.39 g kg^−1^).

In MS, the concentration of Ca was higher in CS-M08sh, CS-T02, and CS-T04, with CS-M08sh being the highest (Table 3). Potassium concentration was higher in CS-M02 and AM UA48 compared to TM-1. CS-T04 and CS-T08sh had higher Mg concentrations than the controls and the other CS lines. Phosphorus concentration was higher in four CS lines (CS-M04, CS-M08sh, CS-T02, and CS-T04) than the controls, with CS-T04 and CS-T08sh being the highest. Carbon concentration was higher in the controls, with TM-1 being higher than all CS lines. Three (CS-B08sh, CS-M08sh, and CS-T02) CS lines exhibited higher N content than the controls, with both CS-B08sh and CS-T02 being the highest. Seven (CS-B02, CS-B04, CS-B08sh, CS-M04, CS-M08sh, CS-T02, and CS-T08sh) lines showed higher S concentration than the controls, with CS-T08sh being the highest.

### 2.1. CS Lines with High Level of Nutrients in Both Locations

Two CS lines (CS-T02 and CS-T04) showed higher Ca in both locations compared to the controls. CS-T04 had higher Mg concentrations in both locations. Three CS lines (CS-T02, CS-T04, and CS-T08sh) showed superiority over TM-1, but only CS-T02 and CS-T04 showed superiority over both controls. Cultivar AM UA48 was the best performer for C concentration in both locations. CS-T02 exhibited a higher N concentration than the controls. Four CS lines (CS-B02, CS-B04, CS-T02, and CS-T08sh) exhibited a higher concentration of S than the controls.

### 2.2. Correlation and Distribution

In SC, P positively correlated with Ca and Mg, N positively correlated with Ca and K, and S positively correlated with Ca, Mg, P, and N. In MS, P positively correlated with Ca and Mg, N positively correlated with Mg and negatively with P, and S negatively correlated with P and positively with N (Table 4). It must be noted that some nutrients showed positive and significant (*p* ≤ 0.0001) correlation in both locations; for example, P positively correlated with Ca, P positively correlated with Mg, S positively correlated with P, and S positively correlated with N (Table 4 and Table 5). In both locations, the highest positive correlation was detected between P and Mg, explaining about 45% and 32% of the variation, respectively, in SC and MS. A wide nutrient distribution and significant variability for all macronutrients, especially Ca, K, N, and S, across all CS lines and between CS lines and controls (Figure 1 and Figure 2) signify the potential use of CS lines with high levels of these nutrients for future breeding programs to increase the nutritional quality of cottonseed.

## 3. Discussion

Since cotton chromosome substitution lines are near-isogenic to the recurrent parent TM-1 with one chromosome or chromosome arm introgressed from *G. barbadense* 3−79, differences in macronutrient levels in seeds should be mainly due to the introgressed chromosome or chromosome arm. Although the association of seed traits with chromosomes or chromosome arms has been previously used [26,27,28,29,30,31], the effects of chromosome substitution germplasm lines on macronutrients levels in cottonseed is non-existent and what is available in the literature is research conducted to identify other desirable cottonseed traits. For example, the effects of chromosome substitution on seed protein and oil were studied in F_3_ hybrids of 13 cotton chromosome substitution lines crossed with five elite cultivars grown in four environments [32]. In this study, they observed significant additive effects for seed protein and oil content and concluded that chromosome associations (specific chromosomes or chromosome arms in the TM-1 background), using comparative methods, helped to improve seed nutritional quality traits [32,33]. In another study conducted to identify desirable seed traits in five commercial cultivars, 13 CS lines, 3−79 (donor parent), TM-1 (recurrent parent), and F_3_ hybrids [29,30], the authors used chromosome association (comparative method) with seed traits and concluded that using chromosome substitution germplasm lines should provide valuable genetic information for desirable seed trait analysis in other crops. In the current study, the comparative approach showed that some CS lines had higher macronutrients in seeds than others in SC or MS, suggesting the positive effects of these chromosomes or chromosome arms on these nutrients. For example, in SC, the CS lines CS-T02, CS-T04, CS-T08sh, CS-B02, and CS-B04 had higher Ca levels in seed compared with the controls; CS M08sh showed the highest K concentration (14.50 g kg^−1^). A similar observation was noticed for lines CS-B02, CS-B04, CS-B08sh, CS-M02, CS-M04, CS-M08sh, CS-T02, and CS-T04 for Mg concentration; lines CS-M04 and CS-T04 for P concentration; and lines CS-B08sh, CS-T02, CS-T04, and CS-T08sh for N concentration. Other CS lines showed negative responses for some macronutrients; for example, CS-M02 and CS-M04 showed reduced N compared to the controls.

In MS, similar responses for macronutrients in CS lines were shown when compared to the controls, exhibiting positive, negative, or non-responsive effects of chromosome or chromosome arm substitution as in the SC location. The higher concentration of some nutrients in some CS lines reinforced our explanation that the accumulation of seed nutrients in CS lines were due to introgressed chromosomes as these CS lines are near-isogenic and the only differences between these CS lines and the parent or control are the chromosomes or chromosome arms under study. For example, Ca concentration was high in lines CS-M08sh, CS-T02, and CS-T04 compared with the controls. A similar observation was recorded for K in CS-M02 compared to TM-1; CS-T04 and CS-T08sh for Mg compared with the controls; lines CS-M04, CS-M08sh, CS-T02, and CS-T04 for P compared with the controls; lines CS-B08sh, CS-M08sh, and CS-T02 for N compared with the controls; and lines CS-B02, CS-B04, CS-B08sh, CS-M04, CS-M08sh, CS-T02, CS-T08sh for S content. The negative response of nutrients in some CS lines was shown in, for example, CS-B02, CS-B04, and CS-M04 for C. It must be noted here that some seed nutrients showed different location responses in some CS lines, reflecting the significant effects of environmental conditions at each location. Previous research on other seed traits also showed different responses, depending on the environmental factors of each location, including soil type, temperature, drought, and other environmental conditions [13]. For example, during the critical months of growth (May through September), the temperature was higher in Mississippi state (Starkville, MS) than in Florence, SC, and precipitation was higher in Mississippi state (Starkville), except in July. These differences could be possible sources of variability. The differences in temperature and rainfall could affect nutrient uptake, translocation, and accumulation in cottonseed at different levels depending on their genotypic adaptability to environmental conditions, soil type, and other growing conditions. On the other hand, some nutrients in CS lines showed higher contents in both locations, indicating the possible stability of these traits across location/environment. For example, lines CS-T02 and CS-T04 showed higher Ca in both locations, and CS-T04 had higher Mg concentrations in both locations. A similar observation was recorded in lines CS-T02 and CS-T04 for P; CS-T02 for N; and CS-B02, CS-B04, CS-T02, and CS-T08sh for S.

Among the macronutrients evaluated in the CS lines, some showed a positive response at one location, while others showed a negative response at the same location, and others showed a positive response in both locations. Previous research conducted on other seed traits found similar observations. For example, ref. [32] found homozygous and heterozygous dominance effects and additive effects on cottonseed protein and oil. They reported that 9 out of 20 parents showed significant negative homozygous dominance effects for oil content, which was positive for four parents. It was explained that chromosomes 2, 7, 18, 25, and chromosome arm 5sh of the 3−79 donor parent were associated with high seed oil content cotton variety DP90, indicating the positive heterozygous dominance effects; however, chromosome arms 22sh and 22Lo were associated with low seed oil content, reflecting the negative heterozygous dominance effects [32]. SG747 showed a similar response in that the heterozygous dominance effects for chromosomes 2 and 17, and chromosome arms 15sh, 22sh, and 22Lo were associated with higher seed oil content, while chromosomes 16 and 25 were associated with low oil content [32]. Other studies using a RIL population showed that mineral nutrient concentrations were significantly improved compared to the parents and concluded that genetic diversity for mineral nutrient concentration can provide novel alleles for higher grain mineral concentrations [34]. In our study, the results showed that some CS lines exhibited increased Ca, K, Mg, P, N, and S; while others showed a decrease in some of these nutrients.

A similar observation was noted at the Mississippi location, with different responses of the CS lines due to location effects. These observations demonstrated that chromosome substitution resulted in a higher concentration of macronutrients in some CS lines, with no change or reduced concentrations in others. The positive or negative response of these macronutrients was due to both location and CS line genotype, as explained by others above [32,34]. The CS lines with high concentrations of macronutrients in both locations (CS-T02 and CS-T04) for Ca; CS-T04 for Mg; CS-T02 and CS-T04 for P; CS-T02 for N; and CS-B02, CS-B04, CS-T02, and CS-T08sh for S can be used as parents in a breeding program for selection to increase cottonseed nutritional qualities by crossing these desirable CS lines to generate elite germplasm and cultivars. Previous research has reported that the genetic system for cottonseed traits is complicated and multiple genetic models are involved [35,36,37], but these complicated genetic models can be useful to identify desirable cottonseed traits [32,37,38]. The identification of these CS lines with high macronutrients in seeds compared with both controls in both locations indicated increased nutrient uptake efficiency by these CS lines that may overcome deficiencies of these nutrients under soil nutrient deficiency conditions.

The positive correlation of P with Ca and Mg; N with Ca and K; S with Ca, Mg, P, and N in SC; the positive correlation of P with Ca and Mg; N with Mg; the negative correlation of N with P; and the negative correlation of S with P and positive correlation with N in MS indicated that the correlation between some nutrients can be influenced by complex genetic and environmental factors [39]. The consistently significant (*p* ≤ 0.0001) positive correlation between P and Ca, P and Mg, S and P, and S and N in both locations may suggest the stability of these traits across locations, and these traits may help growers for fertilizer management and advance our knowledge of nutrient uptake and translocation. The wide distribution, especially in Ca, K, N, and S, and significant variability of nutrients among CS lines and between CS lines and the controls signified the potential use of CS lines with high levels of these nutrients for future breeding programs to increase the nutritional quality of cottonseed. The right skewed, left skewed, or peak trends, shown by the nutrient distribution between CS lines, showed complex interactions of these nutrients in these lines.

This research demonstrated that the improvement of cottonseed nutritional quality using chromosomes or chromosome segments from the wild species *G. tomentosum* and *G. mustelinum* is possible. This is the first report on the effects of chromosome or chromosome segment substitution on macronutrients in cottonseed. We were able to discover specific chromosomal associations with cottonseed macronutrient contents gene(s) and developed some novel germplasm with the gene(s) of improved macronutrient contents from interspecific crosses using the wild species.

## 4. Materials and Methods

### 4.1. Chromosome Substitution Cotton Lines (CS)

Nine cotton chromosome substitution lines were used and represented by the substitution of chromosome two, four, and the short arm of chromosome eight (8sh) from three tetraploid species of *G. barbadense* L. (CS-B), *G. tomentosum* Nutt.Ex Seem (CS-T), and *G. mustelinum* Watt (CS-M), respectively, into Upland cotton (TM-1, *G. hirsutum* L.), the recurrent parent of the CS lines, and AM UA48, Reg. No. CV−129, (Bourland and Jones, 2012). The nine CS lines were: CS-B02, CS-B04, CS-B08sh, CS-M02, CS-M04, CS-M08sh, CS-T02, CS-T04, CS-T08sh; TM-1 and AM UA48 were used as controls. The developed lines are described in detail elsewhere [[26][27][28][29][30],[40]]. The lines were planted under field conditions in two locations: in Florence, South Carolina (SC) (34.1°N, 79.4°W) in 2013; and in Mississippi State, Starkville (MS) (33.4°N, 88.8°W) in 2014. The SC soil type was a Norfolk loamy sand (fine-loamy, kaolinitic, thermic typic Kandiudults). The MS soil type was a Leeper silty clay loam (fine, smectitic, nonacid, thermic Vertic Epiaquept). The soil type, field conditions, and planting were previously described by others [40]. Each line was grown in a single-row, 12 m long, with rows spaced 97 cm apart and plants spaced 10 cm apart (about 110 plants per row). Seeds from 25 open-pollinated-boll from each line were hand-harvested at both locations from the first fruit positions near the middle nodes of the plants to determine cottonseed mineral analysis. Samples were ginned on a 10-saw laboratory gin to separate seeds from the lint. Seeds from these samples were de-linted and analyzed for mineral analysis.

### 4.2. Seed Mineral Nutrient Analyses

The macronutrients Ca, K, Mg, P, C, N, and S were analyzed in the ground, dried whole seed samples. Collected seeds from each CS line and control were ground with a Laboratory Mill 3600 (Perten, Springfield, IL, USA), and analyzed by digesting 0.5 g of dried ground seed in HNO_3_ in a microwave digestion system. The concentrations of nutrients were determined by inductively coupled plasma spectrometry (ICP) (Thermo Jarrell-Ash Model 61E ICP and Thermo Jarrell-Ash Autosampler 300) [41]. The nutrients C, N, and S were determined from 0.25 g samples using a C/N/S elemental analyzer as detailed previously [41,42]. Seed P was analyzed as detailed below.

### 4.3. Determination of Seed P

The concentration of P in cottonseed was determined by the yellow phosphor vanado–molybdate complex method [43], and as previously described [41,42]. Phosphorus was extracted with 2 mL of 36% *v/v* HCl. Then, 5 mL of 5 M HCl and 5 mL of ammonium molybdate–ammonium metavanadate was used. The concentration of P was determined by a Beckman Coulter DU 800 spectrophotometer as previously described by others [41,42]. The absorbance was read at 400 nm.

### 4.4. Experimental Design and Statistical Analysis

Cotton lines were grown in a randomized complete block design with four replications within each location. Proc Mixed (SAS, SAS Institute, 2002–2010) [44] was conducted to evaluate the effects of location, line, and their interactions. Replicates within a location were considered as random effects. Line and location were considered as fixed effects. Multiple comparison procedures (mean separation test) were conducted at a significance level of 5% in SAS [44]. Since location by line interactions were significant for macronutrients, results were presented separately by location. Significant differences in nutrients between a specific CS line and TM-1 should be due to the specific substituted chromosome or chromosome arm from the donor parent of *G. barbadense*, *G. tomentosum,* and *G. mustelinum,* because an individual CS line is considered to be near-isogenic in a background of TM-1. Correlation analyses were conducted in SAS [44] using Proc Corr.

## 5. Conclusions

The current research demonstrated the possibility of increasing major macronutrient (Ca, K, Mg, P, and S) levels in cotton using wild species germplasm introgression. In this research, we used a comparative method [32] to detect potential effects of disomic chromosome substitution on seed macronutrients. CS lines that exhibited higher macronutrients in both locations might be of interest as parents for genetic dissection and breeding for seeds with higher macronutrient content, increasing cottonseed meal nutritional quality in its use for human nutrition and livestock feed. For example, lines CS-T02 and CS-T04 showed higher Ca in both locations than the controls; CS-T04 had higher Mg concentrations than the controls in both locations; lines CS-T02, CS-T04, and CS-T08sh showed superiority to TM-1; line CS-T02 and CS-T04 showed superiority to the controls; CS-T02 exhibited higher N concentrations than the controls; and lines CS-B02, CS-B04, CS-T02, and CS-T08sh had a higher concentration of S than the controls. The wide range of nutrients among the CS lines is important in breeding selection to obtain cottonseed lines with higher nutritional qualities. The cotton lines with higher cottonseed nutrient levels can be used as a source for human nutrition and livestock feed. The different responses of macronutrients to chromosome substitution in different CS lines indicated that certain chromosomes of the TM-1 genome were associated with increased macronutrient levels, other chromosomes were associated with decreased macronutrients, and others did not change the level of macronutrients compared with TM-1. The results reported in the current study, to our knowledge, are the first to report on the effects of chromosome substitution on cottonseed macronutrients. Further research is needed to check the stability of chromosome effects on seed nutrients across years and locations as the current research was conducted in two locations only, but not across multiple years.

## Figures and Tables

**Figure 1 plants-10-01158-f001:**
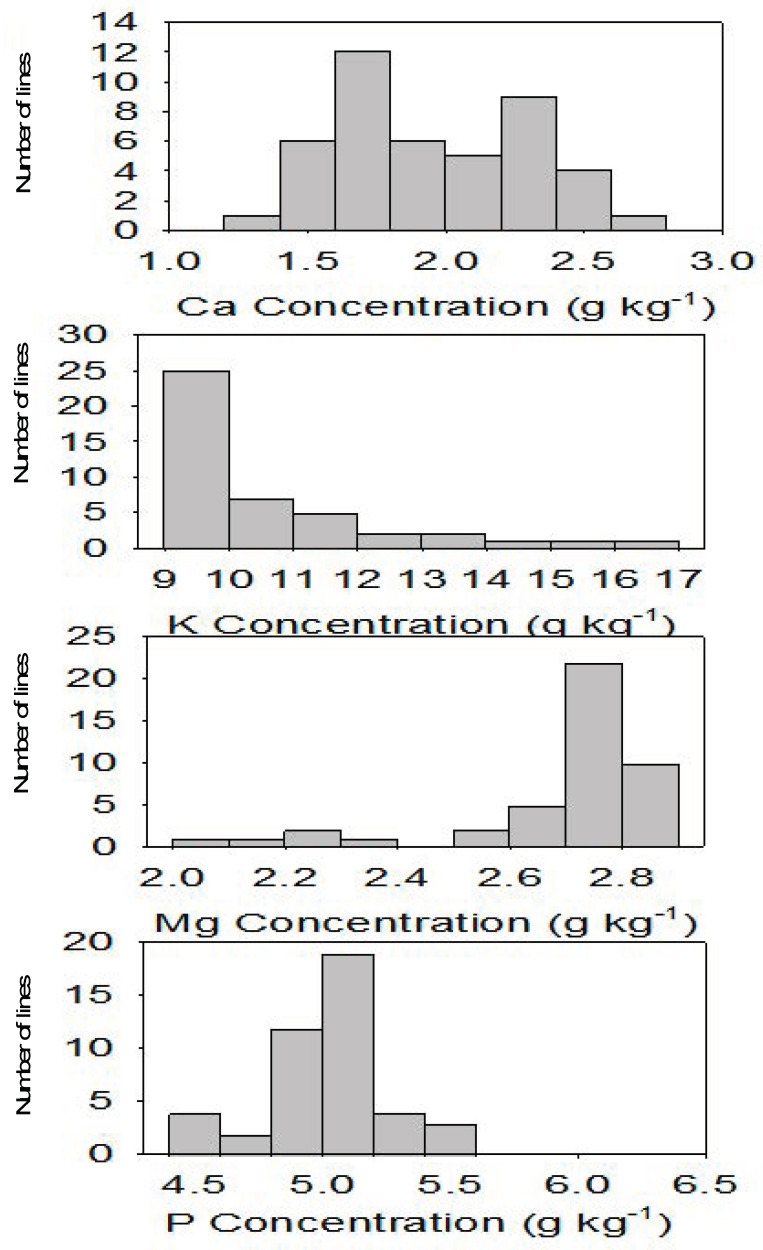
Distribution of macronutrients from top to bottom, respectively, for Ca, K, Mg, and P in cottonseed chromosome substitution lines (CS) across locations. X-axis is nutrient concentration; Y-axis is number of lines.

**Figure 2 plants-10-01158-f002:**
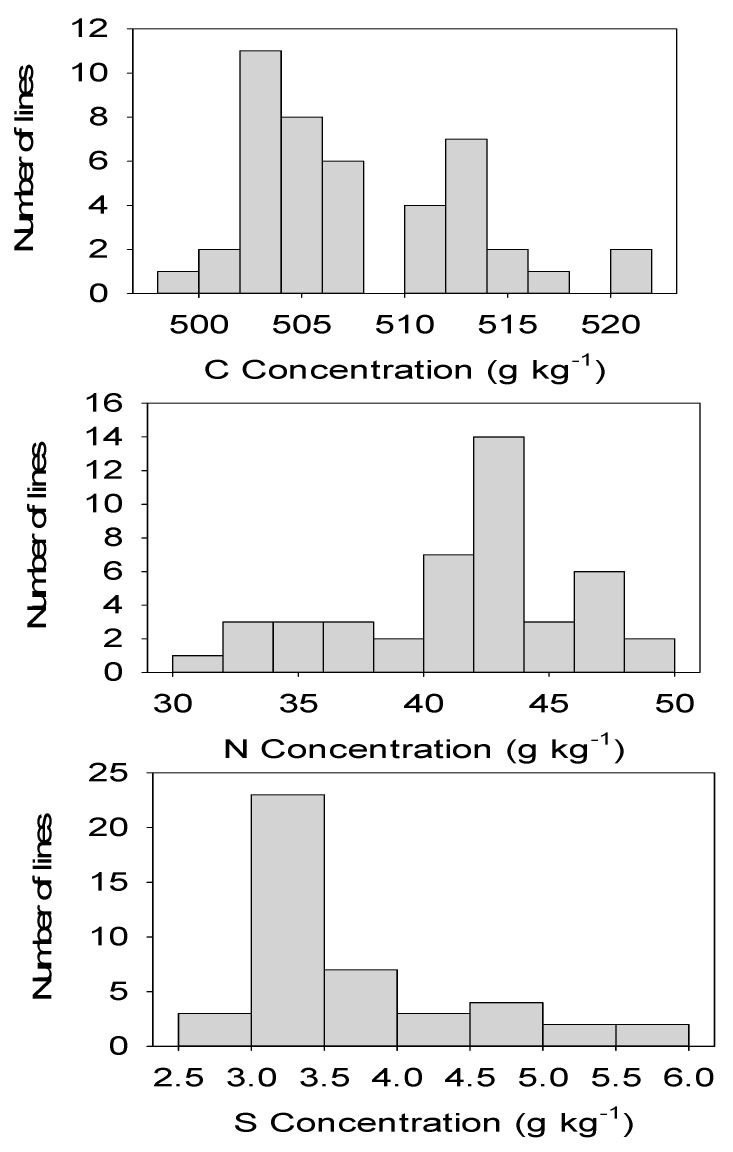
Distribution of macronutrients from top to bottom, respectively, for C, N, and S in cottonseed chromosome substitution lines (CS) across locations. X-axis is nutrient concentration; Y-axis is number of lines.

**Table 1 plants-10-01158-t001:** Analysis of variance (F and *p* values) of the effects of location, genotype (CS lines, parent TM-1, and commercial cultivar AM UA48), and their interactions for cottonseed minerals (mg kg^−1^) in cotton chromosome substitution lines (CS) in different locations (Florence, SC, 2013 and Starkville, MS, 2014).

Effect	DF	Ca		K		Mg		P		C		N		S	
		F	*p*	F	*p*	F	*p*	F	*p*	F	*p*	F	*p*	F	*p*
Location	1	3.86	ns	2.98	ns	139.81	***	494.42	***	1.46	ns	22.68	***	22.51	***
Line	10	40.63	ns	6.49	***	5.32	***	18.96	***	1.23	ns	36.95	***	31.64	***
Location × line	10	26.49	ns	8.44	***	8.81	***	15.43	***	1.25	ns	21.11	***	32.18	***
Residual		0.011		0.585		0.016		0.038		22.7		3.069		0.046	

DF = degree of freedom. *** significance at *p* ≤ 0.001.

**Table 2 plants-10-01158-t002:** Experiment 1: Effects of cotton chromosome substitution (CS) on cottonseed macronutrient concentrations (g kg^−1^). The experiment was conducted in 2013 in Florence, SC, USA.

Line	Ca	K	Mg	P	C	N	S
CS-B02	2.04D	9.69D	2.81AB	5.07DE	507CD	42.26CD	4.33B
CS-B04	1.88E	9.65D	2.73D	5.05E	508BC	42.65C	3.31D
CS-B08sh	1.80F	10.16C	2.78DCD	5.10CD	505DE	44.02B	3.44C
CS-M02	1.49I	9.62D	2.78BCD	4.82F	508BC	35.55E	3.50C
CS-M04	2.14G	10.96B	2.75BCD	5.15B	506CDE	35.28E	3.18E
CS-M08sh	1.64H	14.50A	2.74CD	5.08DE	511A	34.00F	3.26DE
CS-T02	2.63A	9.67D	2.80ABC	5.12BC	503E	47.79A	4.26B
CS-T04	2.30B	10.41C	2.85A	5.38A	506CD	44.09B	5.39A
CS-T08sh	2.18C	11.21B	2.57E	5.11CD	511AB	43.75B	3.31D
TM-1	1.81F	11.25B	2.21F	4.48G	504DE	41.92CD	3.17E
AM UA48	1.72G	9.74D	2.55E	5.10CD	512A	41.67D	2.89F

Means within a column followed by the same letter are not significantly different at the 5% level.

**Table 3 plants-10-01158-t003:** Experiment 2: Effects of cotton chromosome substitution (CS) on cottonseed macronutrient concentrations (g kg^−1^). The experiment was conducted in 2014 in Mississippi State (Starkville, MS, USA).

Line	Ca	K	Mg	P	C	N	S
CS-B02	2.13C	9.95 DE	3.08 BC	6.11 C	503 G	41.19 D	3.38 CD
CS-B04	2.00D	10.12 CDE	2.94 D	5.51 F	505 DE	35.59 E	3.46 BC
CS-B08sh	1.70F	10.48 BC	2.98 D	5.43 F	507 BCD	45.75 A	4.65 A
CS-M02	1.95D	10.95 AB	2.83 E	5.95 D	507 BCD	34.00 G	3.21 EF
CS-M04	1.88E	10.33 CDE	3.04 C	6.25 B	503 FG	33.25 H	3.46 BC
CS-M08sh	2.55A	10.16 CDE	2.96 D	5.73 E	507 BCD	45.00 B	3.50 B
CS-T02	2.19B	10.25 CDE	2.80 E	6.25 B	506 CDE	45.91 A	3.46 BC
CS-T04	2.24B	10.08 DE	3.21 A	6.68 A	507 BCD	34.75 F	3.30 DE
CS-T08sh	1.98D	9.86 E	3.27 A	6.63 A	505 EF	34.50 FG	2.84 G
TM-1	2.09C	10.29 CDE	3.13 B	6.10 C	511 A	42.47 C	3.14 F
AM UA48	1.51G	10.86 AB	2.85 E	4.95 G	508 B	41.02 D	3.25 E

Means within a column followed by the same letter are not significantly different at the 5% level.

**Table 4 plants-10-01158-t004:** Pearson correlation coefficients (*p* and R values) between cottonseed minerals in cotton chromosome substitution lines (CS) grown in SC 2013.

	Ca	K	Mg	P	C	N
**K**	R = −0.191					
	*p* = ns					
**Mg**	R = 0.190	−0.188				
	*p* = ns	ns				
**P**	R = 0.441	−0.071	0.668			
	*p* = ***	ns	***			
**C**	R = −0.237	0.152	−0.208	0.045		
	*p*= ns	ns	ns	ns		
**N**	R = 0.560	−0.430	−0.074	0.119	−0.043	
	*p* = ***	***	ns	ns	ns	
**S**	R = 0.546	−0.213	0.410	0.436	−0.190	0.401
	*p* = ***	ns	***	***	ns	***

*** significance at *p* ≤ 0.001; ns = not significant at 0.05 probability.

**Table 5 plants-10-01158-t005:** Pearson correlation coefficients (*p* and R values) between cottonseed minerals in cotton chromosome substitution lines (CS) grown in MS 2014.

Ca	Ca	K	Mg	P	C	N
**K**	R = −0.262					
	*p* = ns					
**Mg**	R = 0.191	−0.125				
	*p* = ns	ns				
**P**	R = 0.414	−0.196	0.569			
	*p* = ***	ns	***			
**C**	R = −0.052	0.053	−0.109	−0.070		
	*p* = ns	ns	ns	ns		
**N**	R = 0.146	0.057	−0.294	−0.340	0.050	
	*p* = ns	ns	*	*	ns	
**S**	R = −0.169	−0.011	−0.136	−0.356	−0.138	0.450
	*p* = ns	ns	ns	*	ns	***

* Significance at *p* ≤ 0.05; *** significance at *p* ≤ 0.001; ns = not significant at 0.05 probability.

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
