# Peer review of "Effects of Interspecific Chromosome Substitution in Upland Cotton on Cottonseed Macronutrients"

_plants, 2021, doi:10.3390/plants10061158_

Round 1
Reviewer 1 Report
Dear authors of manuscript entitled „Effects of interspecific chromosome substitution in upland cotton on cottonseed macronutrients”,
Below please find my comments and suggestions related with your work:
INTRODUCTION:
Please explain what relationship exists between describing macronutrients in cotton and macronutrients in human diet? I understand that macronutrients play a key role in the proper functioning of the human body. The authors describe the processes that require these macronutrients, indicate the effects of the deficit of selected macro and micro components. However, it is difficult to find a relationship between this description and the research issues undertaken in the work, which, however, concerns cotton.
L 81-82, authors wrote that “Nutrient accumulation in seeds is controlled by several processes, including nutrients uptake, translocation, redistribution, and accumulation [11,12]. Genetic control of these processes is still not well understood [13]” and then, authors placed citation from article published in 2010. I suppose that from that time, the situation could be changed, and a little bit more of knowledge perhaps have been explored during this 11 years. If I am wrong and there is still no information about this genetic mechanism, please provide evidence from at least 2020.
L89-91, Please explain briefly why the use of QTL has some limitations.
MATERIALS AND METHODS
L306, “briefly” in this place in unnecessary word.
Please indicate in details the origin of the lines used and provide literature references describing the lines used.
RESULTS
I propose to present the obtained results in detail first, and then to discuss their statistical analysis.
L110, The authors refer to Table 2 and write about the content in seeds, but from Table 2 it does not follow that this content is related to the seeds. Please explain.
My general impression is that the results are briefly described and their presentation makes it difficult to understand and evaluate the overall results, for example, figures 1 and 2 are not well described and explained at all.
DISCUSSION
At the beginning of the manuscript, the authors discuss the role of soil in the absorption of micro and macro nutrients by the plant.
Please explain what influence the soil composition, soil type, content of appropriate components, climate, growing conditions, etc. had on the results obtained.
Please describe the influence of the external environment on the obtained results.
Author Response
Please see attachment for our response to Reviewer 1 and Reviewer 2. All editings (revision/adding/modification, and etc.) are track changes or highlighted.

Reviewer 2 Report
This study aims on the evaluation of the effects of chromosome or chromosome arm substitution on cottonseed macronutrients of upland cotton. Current research has been focused on the utilization of different germplasm lines with desirable seed traits. The authors published recently a similar paper as well.
Although these practices are useful in order to improve crop quality under environmental constraint, this paper is quite poor concerning the experimental method and data presentation.
Specific comments:
Abstract: Too long too much information, it should be shortened.
Introduction: The uses of cottonseed in human nutrition should be referred, especially those of macronutrients studied.
Materials & Methods: Define if the experiment has been done in growth chamber or in the field. Some information about the soil and environmental conditions must be written. Experiments took place 7-8 years ago!!
Results: I do not see the analysis of Figs 1 and 2 all through the text.
Discussion: Which is the meaning of higher Ca, Mg, P, K, C concentration in cottonseed for human nutrition? Are they desirable indeed? I suggest these results to be combined with other desirable traits eg lipids, proteins, gossypol etc.
The paper can be accepted in “Plants” after major revision following my recommendations.
Recommendation
Major Revision
Author Response

(The authors gave the same response as above.)

Round 2
Reviewer 1 Report
Dear Authors, please check the figure 1, I can not see the result on the charts in my pdf. version.
Reviewer 2 Report
Since the authors have made the appropriate changes to the revised Ms following my suggested comments, I think that the paper can be accepted in its present form.
